# Examining and Improving the Gender and Language DIF in the VERA 8 Tests

Güler Yavuz Temel *, Christian Rietz, Maya Machunsky  and Regina Bedersdorfer

Faculty of Educational and Social Sciences, University of Education Heidelberg, 69120 Heidelberg, Germany; christian.rietz@ph-heidelberg.de (C.R.); machunsky@ph-heidelberg.de (M.M.); bedersdorfer@ph-heidelberg.de (R.B.)
* Correspondence: yavuz@ph-heidelberg.de

**Abstract:** The purpose of this study was to examine and improve differential item functioning (DIF) across gender and language groups in the VERA 8 tests. We used multigroup concurrent calibration with full and partial invariance based on the Rasch and two-parameter logistic (2PL) models, and classified students into proficiency levels based on their test scores and previously defined cut scores. The results indicated that some items showed gender- and language-specific DIF when using the Rasch model, but we did not detect large misfit items (suspected as DIF) when using the 2PL model. When the item parameters were estimated using the 2PL model with partial invariance assumption (PI-2PL), only small or negligible misfit items were found in the overall tests for both groups. It is argued in this study that the 2PL model should be preferred because both of its approaches provided less bias. However, especially in the presence of unweighted sample sizes of German and non-German students, the non-German students had the highest misfit item proportions. Although the items with medium or small misfit did not have a significant effect on the scores and performance classifications, the items with large misfit changed the proportions of students at the highest and lowest performance levels.

**Keywords:** IRT; RMSD and DIF; item misfit; VERA 8; proficiency levels

## 1. Introduction

Differential item functioning (DIF) can be a typical source of item misfit [1], and it occurs as a difference in item parameters across different subgroups (e.g., male and female), despite controlling for the underlying ability [2,3]. It can manifest at the following levels: that of individual items, known as DIF [4]; that of the overall score, known as differential test functioning (DTF); or at both levels, known as differential functioning of items and tests (DFIT) [5]. During the past few decades, there have been many analyses of DIF based on a growing number of DIF statistics and procedures, presenting different reasons why DIF analysis should be a routine part of the development of cognitive assessments.

Large-scale assessments (LSAs), such as the Programme for International Student Assessment (PISA), prioritize tests that yield comparable scores for the comparison groups. Students from various groups take the test, including those from diverse linguistic, ethnic, and socioeconomic backgrounds [6]. Many studies have focused on examining measurement comparability for different language or gender groups in a cognitive assessment or a questionnaire. One of the important methodological challenges is the non-invariant item parameters across countries, and this national DIF has been extensively studied in the literature [7–10].

In the PISA, different calibration methods have been used to obtain more stable and less biased estimates. Concurrent calibration under full invariance was used until PISA 2012, and since PISA 2015 a concurrent calibration under partial invariance has been established [11]. The concurrent scaling under partial invariance has been used in DIF statistics and, in contrast to concurrent scaling under full invariance, concurrent scaling

under partial invariance allows some of the item parameters with large DIF effects to vary across groups, and the DIF effect is not ignored [11]. Especially in recent years, partial invariance has been often proposed for valid comparisons [12–16]. On the other hand, there is an important argument about the removal of the construct's relevant items with DIF, and researchers have argued that removing this item may lead to construct underrepresentation [17,18]. Similarly, it was emphasized that there is no direct route from statistical bias to unfairness [19], and a simulation study [20] showed that unbiased estimates of national means can be obtained if noninvariance does hold (i.e., all items have DIF effects, but are anchor items). In addition, different linking methods were compared in different test conditions and simulation studies [11], and the results of these studies showed that in the case of unbalanced DIF effects, robust linking methods are preferred over non-robust alternatives, while for balanced DIF all linking methods produced unbiased estimates [20].

One of the important issues with PISA scaling is the choice of the IRT model. Researchers have argued that better-fitted IRT models with partial invariance can lead to valid comparison of countries in the PISA [13,14]. However, on the other hand, a recent methodological case study showed that concurrent calibration under full invariance and partial invariance resulted in very similar national means [21]. For instance, the impact of the choice of the IRT model on the distribution parameters of countries was investigated in a study [19], and the variability in model uncertainty was quantified using the model error, with the results showing that the model uncertainty was relatively small compared to the sampling error in terms of national means in most cases, but was substantial for national standard deviations and percentiles, demonstrating that uncertainty regarding the IRT scaling model influences national means. In another study, it was argued that the search for the correct model should be abandoned, and the authors allowed the regression model to be incorrect and provided estimators that were asymptotically unbiased and standard errors that were asymptotically correct even when there were important specification errors [22]. Similarly, it has been shown that the ML estimate can be consistent for misspecified models [23].

VERA (Vergleichsarbeiten) is one of the most important performance assessments in Germany. In addition to the different large-scale assessments (such as the Programme for International Student Assessment (PISA) and the Trends in International Mathematics and Science Study (TIMSS)), VERA tests are used by the federal states to inform teachers about the relative performance of their students, classes, and schools.

VERA test scores were compared for different students with diverse gender, academic achievement, main daily language, and socioeconomic background. This heterogeneity can manifest itself in ways that extend beyond differences in proficiency levels. According to various studies and VERA reports [24–27], there is consistent evidence concerning gender differences and language differences. For instance, the students whose mainly daily language is not German (non-German)—who often come from immigrant families and lower socioeconomic status than German families—have lower academic achievement on German language tests [28,29] or on both German and mathematics tests [24,25].

In the evaluation of the VERA 8 reports [24,25], a large proportion of non-German students were classified at low achievement levels on all tests. In addition, female students were classified at high proficiency levels in the language tests (German, English) and subtests (reading, spelling). On the mathematics tests, by contrast, more male students were classified at the highest performance level. Students whose mainly daily language is not German often have more problems in reading and understanding German texts [28]. VERA test scores provide crucial information about the school's or student's development, and are used to improve teaching and learning conditions and make necessary modifications. Teachers use information from VERA to understand the academic needs of the students. Every student is assigned to a proficiency level based on their VERA test score. The reliability and validity of the scores impact these proficiency levels and the decisions by teachers and parents affecting the students' futures [29].

The central focus of this study was to examine gender and language DIF on the VERA reading, spelling, and mathematics tests with different models, and with full and partial invariance approaches.

In this article, we aim to examine gender and language DIF for the VERA 8 Booklet 1 tests (reading, spelling, and mathematics), with full and partial invariance assumption, based on the Rasch model [30] and two-parameter logistic model (2PLM) [31] for dichotomous items. We examined the following research questions through the study:

1.a.  Which items in the VERA tests demonstrate DIF across the gender and language groups?
1.b.  Does using a better-fitted model improve bias items on the subtests and subgroups?
2.  What is the effect of the misfit items on the proficiency classifications?

Detection and Evaluation of DIF

Although many statistical methods for detecting DIF have been described in the literature (e.g., the Mantel–Haenszel (MH) method [32], logistic regression [33], and the item response theory likelihood ratio (IRT-LR) test [34]), in recent years, several alternative methods have been proposed for assessing measurement invariance when many groups and items are included in the analysis, and when the sample size is large [12]. Multigroup concurrent calibration with partial invariance constraints [12–16,35,36] assumes that meaningful comparisons can still be made between groups when there are some violations of scalar invariance that threaten the equality of the measurement model across groups. This method establishes partial invariance across populations in the context of IRT [12].

In this study, we used this approach to investigate measurement invariance, based on item fit with the IRT model [13–16,35–37]. The root-mean-square deviation (RMSD) [38], which has also been defined in the literature [39] as root-mean-square error of approximation (RMSEA), can be regarded as a measure of the magnitude of local misfit, and is sensitive to intergroup variability in both the item location and the item discrimination [12,36].

RMSD is used as an item-fit statistic for examining item parameter invariance in various large-scale assessment studies (e.g., the PISA, the Organization for Economic Cooperation and Development [40], and the Program for the International Assessment of Adult Competencies (PIAAC)). The general approaches (e.g., chi-squared and the likelihood ratio approaches such as Bock's $X^2$ [41], Yen's $Q_1$ [42], Orlando and Thissen's S-$X^2$ [43,44], Wright and Masters' infit and outfit mean-square (MSQ), infit and outfit to statistics [45], Orlando and Thissen's S-$G^2$ [43]) in the literature have inflated type I error rates and lack of power to detect item misfit, especially for large sample sizes [37,46,47]. RMSD can be used as a test statistic that outperforms other item-fit approaches [37] with a large sample size.

For a given latent variable, $\theta$, the RMSD statistic is calculated as follows:

$$\mathrm{RMSD} = \sqrt{\int \left( (P_o(\theta) - P_e(\theta))^2 f(\theta) d\theta \right)}$$

Quantifying the difference between the observed item characteristic curve ((ICC), $P_o(\theta)$) with the model-based ICC ($P_e(\theta)$), weighted by $\theta$ distribution, $P_o(\theta)$ and $P_e(\theta)$ denote the model-based and empirical probability of obtaining a correct response given $\theta$, respectively, while $f(\theta)$ denotes the density function of $\theta$ [36,40,48]. RMSD has values between 0 and 1, and the larger values represent poorer item fit or item misfit. The RMSD cut values for item fit are determined as in the PISA test ([13–15,40], namely, for cognitive scales RMSD > 0.1, and for questionnaire constructs RMSD > 0.3. However, Köhler et al. [37] applied a parametric bootstrapping method and suggested different cut scores with the following definitions of the size of item misfit: RMSD < 0.02 = negligible misfit, $0.02 \leq$ RMSD < 0.05 = small misfit, $0.05 \leq$ RMSD < 0.08 = medium misfit, and RMSD $\geq$ 0.08 = large misfit. Buchholz and Hartig [36] set different cut values and suggested RMSD = 0.055 as a cut value (RMSD = 0.055 (location), RMSD = 0.036 (discrimination)). Following these suggestions, in this study the RMSD values greater than 0.05 (i.e., medium

or large misfit) are accepted as misfits, while the RMSD values greater than 0.1 (large misfit) are suspected as DIF.

## 2. Materials and Methods

### 2.1. The VERA Assessment

VERA (Vergleichsarbeiten) is an important performance test that students take in the 3rd and 8th grades (VERA-3 and VERA-8) in all 16 states of the Federal Republic of Germany. VERA tests are centrally developed at the Institute for Educational Quality Improvement (Institut zur Qualitätsentwicklung im Bildungswesen (IQB)) in Berlin. After test development, including pretesting, the IQB provides the test items to all federal states of Germany. The states then have the option to revise the test; hence, the test results are not necessarily comparable across all German states. In this study, our dataset consists of students from the federal state of Baden-Württemberg in 2019. In 3rd- and 8th-grade classes, VERA assesses student performance in German and mathematics, and in 8th-grade classes, VERA also assesses performance in English and French. Furthermore, VERA tests not only provide overall test scores, but also report scores of the subdomains of a subject. For the subject of German, for instance, listening and reading are reported in VERA 3, and a variety of German spelling and reading tasks are reported in VERA 8. In VERA, the focus is on measurement of cognitive assessment (e.g., reading competence). The Rasch model is used for calibrating and analyzing the VERA tests. After calibration and linking procedures, the logit scale is transformed into an educational standard metric (with mean M = 500, standard deviation SD = 100), and the test scores are compared with some cut values for example, the cut-off and standard values of the reading test are listed in Table 1. At the end of the assessment, every student has different proficiency levels according to their scores.

**Table 1.** Proficiency levels with cut values and standard values of the VERA 3 reading test.

| Levels | Cut Values | Standard Values |
|--------|------------|-----------------|
| I | <390 | Below minimum standard |
| II | 390–464 | Minimum standard |
| III | 465–539 | Normative standard |
| IV | 540–614 | Normative standard "plus" |
| V | ≥615 | Optimal standard |

### 2.2. Datasets

The study was conducted using the VERA 8 test that was administered to 8th grade students in Baden-Württemberg in 2019. VERA assesses language and mathematics competencies using different booklets. As the German secondary school system places students according to their achievement level at the end of primary school, two booklets of the VERA 8 tests are available, with Booklet 1 being recommended for students with lower academic achievement. Students whose main daily language is not German have lower success in the VERA test overall. Because the proportion of non-German students was low in the overall VERA test, we used the datasets from Booklet 1 in this study. We used both German tests, consisting of reading and spelling subtests, along with a mathematics test. Each test consisted of a different number of dichotomously scored items. The number of items on the spelling test was 62, the reading test consisted of 41 items, and the mathematics test included 48 items. The sample size of the German test was 52,291, and the sample size of the mathematics test was 51,908. The tests were applied to different subgroups such as gender (female, male) and language groups (German, non-German). After excluding students with missing data, because symmetrical treatment of groups can be advantageous when evaluating DIF across two groups [49], we used weighted sampling in addition to the overall datasets. The sample size of the groups was presented also in Table 2.

**Table 2.** Sample size of subgroups in the tests.

| Sample Sizes | Tests | Female | | Male | | German | | Non-German | |
|---|---|---|---|---|---|---|---|---|---|
| | | N | % | N | % | N | % | N | % |
| Overall Data | German | 24,154 | 46.20 | 28,134 | 53.80 | 52,291 | 81.85 | 9489 | 18.15 |
| | Mathematics | 23,848 | 45.94 | 28,060 | 54.06 | 44,721 | 86.15 | 7187 | 13.85 |

*2.3. Study Design*

In this study, because the unidimensional Rasch model was used for VERA assessment, we first examined the assumptions of the Rasch model (i.e., unidimensionality, local independence, and monotonicity), and then the overall model–data fit for the Rasch, 2PL, and 3PL IRT models. Although IRT models are very useful and powerful in the calibration and analysis of IRT-based tests, the accuracy of their estimation depends on the model's assumptions. To determine unidimensionality, we evaluated M2* statistics by using the R package mirt [50]. Especially in reading tests, a common passage can be a potential source of local item dependence (LID). LD-X2 [51] and Q3 [42] indices can be used for local independence analysis, and the following rules of thumb apply: standardized LD-X2 indices for local independence between item pairs > 10 [52], or residual correlation of an item pair > 0.20 indicated by Q3 [53], are used as indicators of LID. Local independence analysis was performed using the R package mirt. In addition to unidimensionality and LID, the monotonicity of the item functions was also tested. The monotonicity requirement states that the probability of a correct answer must not decrease over increasing latent variable positions. Violations of monotonicity were detected using z-test statistics and item monotonicity plots provided by the R package mokken [54,55].

Violations of one or more of these assumptions indicate a model misfit [56,57]. In addition to the Rasch model, we used 2PL and 3PL models to obtain a relatively better-fitting model. The Akaike information criterion (AIC) [58] and the Bayesian information criterion (BIC) [59] were used to evaluate the general model–data fit. Following the overall model–data fit and model assumptions, we used RMSD as the item-fit statistic, and determined the proportions of items that did not fit in the reading, spelling, and mathematics tests. This analysis was conducted using the R package TAM [60].

In the study to assess DIF, we initially used the IRT-LR method. However, because the sample size was large, with more items in the tests, and we had an unbalanced sample size for the language groups (i.e., there were 42,802 German and 9489 non-German students in the German tests), almost all items were detected as DIF (the item probability functions of the reading test are listed in the Supplementary Materials, Figures S1 and S2). We used the operational method to examine measurement invariance and item fit in the PISA studies [12,40]. This method detects DIF with simultaneous calibration in multiple groups with partial invariance constraints. Under partial invariance, parameters are not newly estimated in the model, but can be freely estimated by the model across groups [61].

Multigroup IRT modeling [62] was used to handle group differences, and it was extended to examine DIF. DIF analysis compares a focus group and a reference group, where the focus group is typically a minority group, and tends to have a smaller sample size than the reference group [63]. This study used multigroup IRT with RMSD for the evaluation bias items. As with the likelihood test, we started with a non-DIF model in which all item parameters were invariant or equal across groups and fitted the data. In the first step of DIF detection, we used a highly constrained multigroup model (in which all item parameters were invariant, or equal, across groups). We estimated the item parameters using the multigroup IRT Rasch and 2PL models. We then used the RMSD_DIF function to examine DIF across groups by using the R package mirt [50]. The large misfit values (RMSD $\geq$ 0.08) were considered as DIF for each item in each group and test.

RMSD values are used for DIF evaluation, and are assessed as non-ignorable DIF if the RMSD fit is poor; in other words, it is assumed that no measurement invariance can

be established across groups for the item. [36]. In the final step, we used a DIF model that allowed the suspected DIF item parameters (which had poor RMSD fit) to be unequal in their estimates with the 2PL model, and we again evaluated the bias items with the RMSD_DIF function (for more information on this RMSD-based procedure, see [12,36].

In the next step, as described in previous studies [64–66], we assessed the practical effects of the misfitting items on the test scores and proficiency levels of the students. If the misfit was practically significant, it was recommended that a better-fitting (i.e., more general) model be used, or that misfitting items be removed [67]. Removing items from the test can distort the content validity of the measurement, especially if those items represent specific content that is important in representing the overall construct being measured. One of the other issues with removing poorly fitting items is that removing some items may be a disadvantage to some students who answered them correctly. Since item development costs time and money, and in some situations removing items may not be preferable, researchers advise using a better-fitting model [64,66]. In this study, we used concurrent multigroup calibration with full and partial invariance based on the 2PL model, and classified students into performance levels according to their test scores and previously defined cut scores.

The overall analysis of the study was performed using R [68], and we used marginal maximum likelihood (MML) estimates [69] that were obtained using customary expectation maximization for item parameter estimation and a weighted likelihood estimator (WLE) [70] for person parameter estimation. We followed similar estimation techniques to those used for the original VERA estimates. For this reason, we used the MML and WLE estimation techniques for item and person parameters, respectively. We also used internal reliability for assessing the reliability of the test results—called empirical reliability—using the mirt [50] package.

## 3. Results

The model assumptions are shown in Table 3. When the model assumptions (i.e., unidimensionality, local independence, and monotonicity) were assessed, according to M2* statistics, the TLI (reading: 0.923, mathematics: 0.943) and CFI (reading: 0.931, mathematics: 0.945) values of the reading and mathematics tests were above 0.90, and the RMSEA (reading: 0.036, mathematics: 0.034) and SRMR (reading: 0.0318, mathematics: 0.0308) values were below 0.05; these tests contained only a few item pairs (reading test: 1, mathematics test: 4) that showed local dependence. The spelling and German tests contained more item pairs (spelling: 30, German: 45) that showed local dependence, and the SRMR (spelling: 0.0574, German: 0.0436) and RMSEA (spelling: 0.0595, German: 0.0457) were equal to or greater than 0.05, while the fit values were less than 0.90. According to the M2* statistics, the spelling and German tests partially violated the assumptions of the Rasch model. These results also support the use of a better-fitting model.

**Table 3.** Results of Rasch model assumptions: unidimensionality, reliability, local independence, and monotonicity.

| VERA 8 | N [1] | M2* | df | RMSEA | SRMR | TLI | CFI | Cronbach's Alpha | Q3 | Mon. |
|---|---|---|---|---|---|---|---|---|---|---|
| Reading | 41 | 55,469.7 | 779 | 0.037 | 0.032 | 0.928 | 0.931 | 0.86 | 1 | - |
| Spelling | 62 | 340,401.1 | 1829 | 0.060 | 0.057 | 0.789 | 0.795 | 0.88 | 30 | 2 |
| German | 103 | 567,597.3 | 5150 | 0.046 | 0.044 | 0.836 | 0.839 | 0.92 | 45 | 2 |
| Mathematics | 48 | 64,062.6 | 1080 | 0.034 | 0.031 | 0.943 | 0.945 | 0.88 | 4 | - |

[1] N shows the number of items in the tests.

In addition to the overall fit statistics of the model datasets, we assessed the item fit in the tests. The results are presented in Figure 1. The cutoff value of 0.05 was used to indicate DIF items. In the German test, nine items were identified, and the proportion (%) of misfit items in the German test was 8.74%. Figure 1 presents these values for the reading test for three items; the proportion of misfit items in the reading test was 7.32%. The number of misfit items was nine in the spelling test (14.52%), and eight items were found in the

mathematics tests, with the percentage of items in the test at 16.67%. These results reveal that the mathematics tests contained the highest proportion of misfit items. The majority of the items on the tests showed negligible or small misfits.

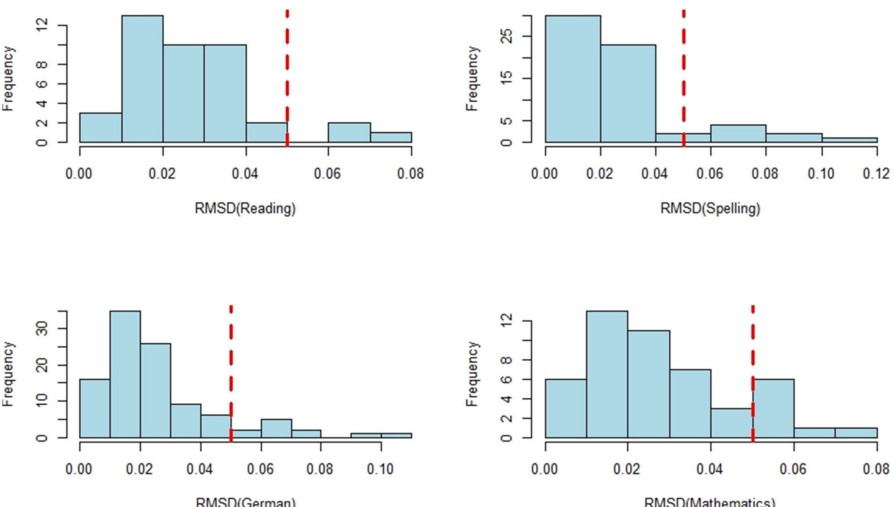

**Figure 1.** Empirical distribution of the RMSD for the binary items on the reading, spelling, mathematics, and German tests. The dotted line indicates the cutoff value of 0.05 used to flag DIF items.

The empirical distribution of the RMSD values of the subgroups is also shown with histograms in Figures 2 and 3. From these graphs, it can also be seen that the mean of the RMSD values was close to 0 and the same for all subgroups in all tests. In addition to these graphs, the means and standard deviations of item parameters (a, d) and abilities were also estimated for each subgroup in the tests with the Rasch and 2PL models. The means and standard deviations of the RMSD values are also reported in Table 4. It was found that the mean of the RMSD values with the Rasch model (0.03) was higher than that of the RMSD values with the 2PL model (0.01) for all subgroups, but with both models the mean values of the subgroups were the same or very close to one another. The mean of the RMSD values for the non-German group was 0.04. The means of the ability parameters were the same for all groups, and the means of these values were equal to 0 while the standard deviations were equal to 1. The means and standard deviations of the discriminant (a) and intercept (d) parameters are also reported in the same table.

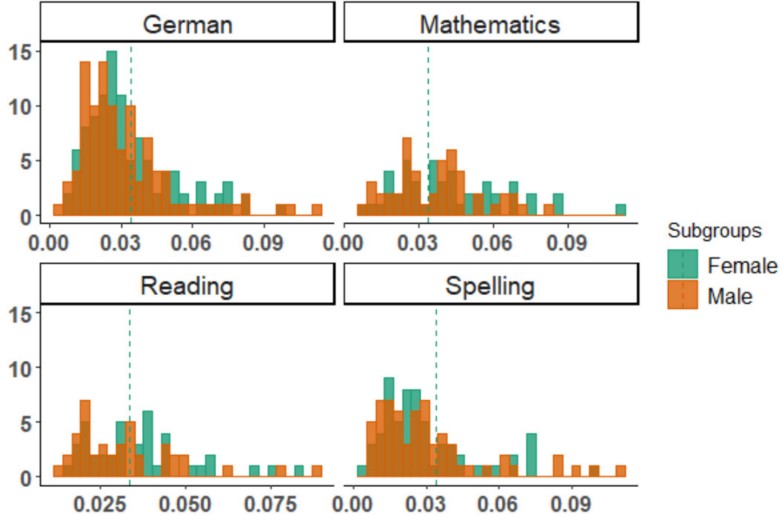

**Figure 2.** Empirical distribution of the RMSD for the items on the reading, spelling, German, and mathematics tests. The dotted line indicates the mean of the RMSD values for gender groups.

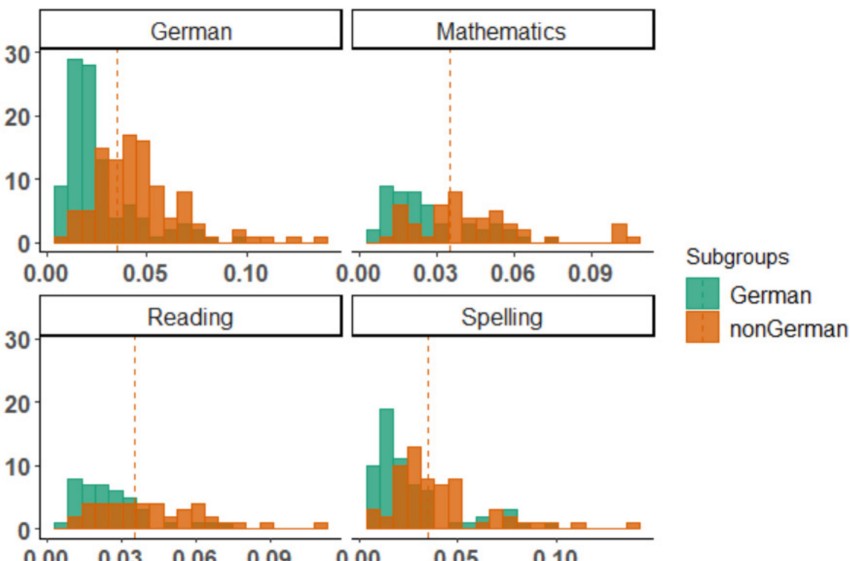

**Figure 3.** Empirical distribution of the RMSD for the items on the reading, spelling, German, and mathematics tests. The dotted line indicates the mean of the RMSD values for language groups.

We also examined gender and language DIF using unidimensional multigroup Rasch, 2PL, and PI-2PL models for the reading, spelling, German, and mathematics tests. Different cut values for assessing misfit items were used (RMSD < 0.02 = negligible misfit, $0.02 \leq$ RMSD < 0.05 = small misfit, $0.05 \leq$ RMSD < 0.08 = medium misfit, and RMSD $\geq$ 0.08 = large misfit). The small and negligible misfit items were assessed with a cut value of 0.03, while the medium and large misfit items were assessed with a cut value of 0.05. Large RMSD values were observed for the gender-specific DIF on each item, and the proportion of misfit items was estimated with Rasch and 2PL models (non-DIF: all item parameters are invariant or equal) as well as with rescaled 2PL (as PI-2PL presented in Tables 5 and 6). The proportions of the small and medium or large misfit items for gender groups are presented in Table 4. We estimated the misfit items with and without sampling weights.

We compared the proportions of misfit items (RMSD > 0.03 and RMSD > 0.05) across the female and male subgroups with three models, and by using the overall dataset and symmetrical sample size of each group. The results showed that the proportion of misfit items was larger in the mathematics test than in the German, reading, and spelling tests. When the results of the small or negligible misfit values (RMSD > 0.03) with the Rasch model were evaluated, the proportion of misfit items was larger for female students than for male students in the reading, German, and mathematics tests. However, in the spelling test, more small or negligible misfit items were identified for the male group. These results were similar to those using the 2PL and PI-2PL models—small proportions of the misfit items were identified for both the male and female groups. Because in the overall dataset the sample size of the female (46.20%) and male (53.80%) groups were not very unbalanced, the results of the small or negligible misfit items with symmetrical sample sizes were similar to those of the non-symmetrical datasets. When we assessed the proportion of the misfit items with a cut value of 0.05 for the non-symmetrical dataset with the Rasch model, the female groups had larger misfit items than the male groups in all tests. The largest proportion of misfit items was estimated with the female group in the mathematics test (33.33%). When the 2PL or PI-2PL models were used instead of the Rasch model, smaller values of medium or large misfit items were estimated in all tests for both female and male students. When we detected misfit items using the PI-2PL model, no medium or large misfit items were detected for either group. For the spelling and German tests, the proportion of misfit items was equal for both subgroups, and in the reading and mathematics tests the female groups

had more misfit items than the male groups, but the difference between them was not very large.

**Table 4.** Results of means and standard deviations of item parameters, abilities, and RMSD values for each group.

| Tests | Groups | | Rasch | | | | 2PL | | | |
|---|---|---|---|---|---|---|---|---|---|---|
| | | | a1 | d | $\theta$ | RMSD | a1 | d | $\theta$ | RMSD |
| Reading | Female | Mean | 1.00 | 0.33 | 0.00 | 0.03 | 0.90 | 0.31 | 0.00 | 0.01 |
| | | SD | 0.00 | 1.07 | 0.96 | 0.02 | 0.29 | 1.06 | 0.96 | 0.01 |
| | Male | Mean | 1.00 | 0.04 | −0.01 | 0.03 | 0.91 | 0.03 | −0.01 | 0.01 |
| | | SD | 0.00 | 0.07 | 0.98 | 0.02 | 0.26 | 1.00 | 1.10 | 0.01 |
| | German | Mean | 1.00 | 0.30 | 0.00 | 0.03 | 0.85 | 0.30 | −0.01 | 0.01 |
| | | SD | 0.00 | 1.02 | 0.93 | 0.01 | 0.24 | 1.01 | 1.11 | 0.01 |
| | Non-German | Mean | 1.00 | −0.45 | −0.01 | 0.03 | 0.93 | −0.47 | −0.01 | 0.01 |
| | | SD | 0.00 | 1.08 | 0.99 | 0.02 | 0.31 | 1.12 | 1.12 | 0.01 |
| Spelling | Female | Mean | 1.00 | 0.78 | 0.00 | 0.03 | 0.85 | 0.81 | 0.00 | 0.01 |
| | | SD | 0.00 | 1.33 | 0.88 | 0.02 | 0.34 | 1.35 | 1.08 | 0.01 |
| | Male | Mean | 1.00 | 0.32 | 0.00 | 0.03 | 0.87 | 0.33 | −0.01 | 0.01 |
| | | SD | 0.00 | 1.27 | 0.87 | 0.02 | 0.31 | 1.28 | 1.08 | 0.01 |
| | German | Mean | 1.00 | 0.61 | 0.00 | 0.03 | 0.86 | 0.64 | 0.00 | 0.01 |
| | | SD | 0.00 | 1.27 | 0.97 | 0.02 | 0.33 | 1.29 | 1.08 | 0.01 |
| | Non-German | Mean | 1.00 | 0.14 | 0.00 | 0.03 | 0.94 | 0.15 | −0.01 | 0.01 |
| | | SD | 0.00 | 1.38 | 0.97 | 0.02 | 0.32 | 1.38 | 1.07 | 0.01 |
| German | Female | Mean | 1.00 | 0.59 | 0.00 | 0.03 | 0.78 | 0.59 | 0.00 | 0.01 |
| | | SD | 0.00 | 1.22 | 0.80 | 0.02 | 0.24 | 1.23 | 1.05 | 0.01 |
| | Male | Mean | 1.00 | 0.21 | 0.00 | 0.03 | 0.79 | 0.20 | 0.00 | 0.01 |
| | | SD | 0.00 | 1.15 | 0.82 | 0.02 | 0.23 | 1.16 | 1.05 | 0.01 |
| | German | Mean | 1.00 | 0.48 | 0.00 | 0.02 | 0.76 | 0.49 | 0.00 | 0.01 |
| | | SD | 0.00 | 1.16 | 0.78 | 0.02 | 0.23 | 1.17 | 1.05 | 0.01 |
| | Non-German | Mean | 1.00 | −0.10 | 0.00 | 0.03 | 0.86 | −0.10 | 0.00 | 0.01 |
| | | SD | 0.00 | 1.27 | 0.87 | 0.02 | 0.26 | 1.29 | 1.05 | 0.01 |
| Mathematics | Female | Mean | 1.00 | 0.15 | 0.00 | 0.03 | 0.97 | 0.16 | 0.00 | 0.01 |
| | | SD | 0.00 | 1.40 | 1.05 | 0.02 | 0.29 | 1.45 | 1.08 | 0.01 |
| | Male | Mean | 1.00 | 0.30 | 0.00 | 0.03 | 1.03 | 0.31 | 0.00 | 0.01 |
| | | SD | 0.00 | 1.36 | 1.07 | 0.02 | 0.25 | 1.40 | 1.08 | 0.01 |
| | German | Mean | 1.00 | 0.24 | 0.00 | 0.03 | 0.89 | 0.22 | 0.01 | 0.01 |
| | | SD | 0.00 | 1.07 | 0.92 | 0.02 | 0.31 | 1.14 | 1.09 | 0.01 |
| | Non-German | Mean | 1.00 | −0.28 | 0.00 | 0.04 | 0.86 | −0.32 | 0.00 | 0.02 |
| | | SD | 0.00 | 1.08 | 0.87 | 0.02 | 0.35 | 1.16 | 1.08 | 0.01 |

**Table 5.** Proportion (%) of misfit items among female and male students on tests.

| Sample | Tests | Groups | RMSD (Cut Value = 0.03) | | | RMSD (Cut Value = 0.05) | | |
|---|---|---|---|---|---|---|---|---|
| | | | **Rasch** | **2PL** | **PI-2PL** | **Rasch** | **2PL** | **PI-2PL** |
| Without Sampling Weights | Reading | Female | 63.42 | 26.83 | 0.00 | 17.07 | 4.88 | 0.00 |
| | | Male | 48.78 | 14.63 | 2.44 | 7.32 | 2.44 | 0.00 |
| | Spelling | Female | 33.87 | 8.07 | 1.61 | 16.13 | 0.00 | 0.00 |
| | | Male | 41.94 | 9.68 | 1.61 | 14.52 | 0.00 | 0.00 |
| | German | Female | 44.44 | 20.37 | 0.93 | 17.48 | 1.94 | 0.00 |
| | | Male | 39.82 | 12.04 | 3.70 | 12.62 | 1.94 | 0.00 |
| | Mathematics | Female | 64.58 | 52.08 | 2.08 | 33.33 | 8.33 | 0.00 |
| | | Male | 58.33 | 31.25 | 2.08 | 18.75 | 6.25 | 0.00 |
| With Sampling Weights | Reading | Female | 63.42 | 24.39 | 0.00 | 14.63 | 0.00 | 0.00 |
| | | Male | 51.22 | 21.95 | 4.88 | 7.32 | 2.44 | 0.00 |
| | Spelling | Female | 33.87 | 9.68 | 1.61 | 14.52 | 0.00 | 0.00 |
| | | Male | 41.94 | 9.68 | 1.61 | 14.52 | 0.00 | 0.00 |
| | German | Female | 42.59 | 19.44 | 7.41 | 17.60 | 1.85 | 0.00 |
| | | Male | 42.59 | 14.82 | 5.56 | 12.96 | 1.85 | 0.00 |
| | Mathematics | Female | 62.50 | 54.17 | 2.08 | 31.25 | 8.33 | 0.00 |
| | | Male | 64.58 | 43.75 | 0.00 | 18.75 | 6.25 | 0.00 |

**Table 6.** Proportion (%) of misfit items for German and non-German students in the tests.

| Sample | Tests | Groups | RMSD (Cut Value = 0.03) | | | RMSD (Cut Value = 0.05) | | |
|---|---|---|---|---|---|---|---|---|
| | | | **Rasch** | **2PL** | **PI-2PL** | **Rasch** | **2PL** | **PI-2PL** |
| Without Sampling Weights | Reading | German | 29.27 | 0.00 | 0.00 | 7.32 | 0.00 | 0.00 |
| | | Non-German | 65.85 | 29.27 | 2.44 | 31.71 | 4.88 | 0.00 |
| | Spelling | German | 25.81 | 0.00 | 0.00 | 12.90 | 0.00 | 0.00 |
| | | Non-German | 58.07 | 30.65 | 8.07 | 25.81 | 8.07 | 0.00 |
| | German | German | 25.00 | 0.00 | 0.00 | 9.71 | 0.00 | 0.00 |
| | | Non-German | 74.07 | 51.85 | 4.63 | 32.04 | 9.71 | 0.00 |
| | Mathematics | German | 31.25 | 0.00 | 0.00 | 14.58 | 0.00 | 0.00 |
| | | Non-German | 75.00 | 47.92 | 6.25 | 31.25 | 10.42 | 0.00 |
| With Sampling Weights | Reading | German | 41.46 | 2.44 | 0.00 | 9.76 | 0.00 | 0.00 |
| | | Non-German | 56.10 | 17.07 | 2.44 | 26.83 | 0.00 | 0.00 |
| | Spelling | German | 40.32 | 6.45 | 3.23 | 12.90 | 0.00 | 0.00 |
| | | Non-German | 46.77 | 11.29 | 3.23 | 14.52 | 0.00 | 0.00 |
| | German | German | 40.74 | 10.19 | 3.70 | 13.89 | 0.00 | 0.00 |
| | | Non-German | 59.26 | 22.22 | 2.78 | 21.30 | 3.70 | 0.00 |
| | Mathematics | German | 50.00 | 18.75 | 6.25 | 14.58 | 0.00 | 0.00 |
| | | Non-German | 68.75 | 20.83 | 12.50 | 35.42 | 12.50 | 0.00 |

In addition to gender, the differences in the proportions of misfit items for German and non-German students were also evaluated. The results are presented in Table 6. One of the most important differences between the language groups was that non-German students had a non-symmetrical sample size (18.15%), and we detected a large difference between German and non-German subgroups. When we used sampling weighting, the differences between subgroups were not very large, similar to the non-symmetrical sample size analysis. There were no misfit items with the 2PL model for German students, but for non-German students there were medium and large misfit items in the tests. For instance, when we assessed medium and large misfit items (if RMSD > 0.05) with the 2PL model and non-symmetrical sample size, large misfit items were detected for non-German students in the reading test (4.88%), in the spelling test (8.07%), and in the German (9.71%) and mathematics tests (10.42%). When the weighted sampling was used for both language groups, we detected very few misfit items for both groups using the 2PL model, although

some items in the German test (3.70%) and mathematics test (12.50%) were detected as misfit for the non-German students. In addition to these, using the PI-2PL model, any medium or large misfit items were not detected.

The second objective of this study was to examine the practical effects of these medium or large misfit items on test scores and proficiency levels. VERA test scores were evaluated using the five proficiency levels. Each student was categorized into a proficiency level according to their score. The proportion of students at each proficiency level was calculated using the Rasch, 2PL, and PI-2PL models. The results showed that 2PL (non-DIF: all item parameters are invariant or equal) and PI-2PL (DIF: item parameters that are unequal in their estimates with the 2PL model) similarly classified students in terms of proficiency levels. The proportions of female, male, German, and non-German students in the tests are shown in Tables 7 and 8.

**Table 7.** The proportion (%) of the students at each proficiency level, by gender subgroup.

| Tests | Proficiency Levels | Female | | | Male | | |
|-------|-------------------|--------|------|--------|------|------|--------|
| | | Rasch | 2PL | PI-2PL | Rasch | 2PL | PI-2PL |
| Reading | V | 11.32 | 13.84 | 13.76 | 6.71 | 8.73 | 8.45 |
| | IV | 26.34 | 24.35 | 24.59 | 20.88 | 20.20 | 19.66 |
| | III | 30.02 | 29.48 | 29.40 | 29.69 | 29.37 | 29.64 |
| | II | 23.76 | 20.30 | 20.13 | 28.70 | 23.50 | 23.88 |
| | Ib | 5.93 | 7.92 | 7.96 | 8.79 | 10.79 | 10.91 |
| | Ia | 2.63 | 4.11 | 4.16 | 5.23 | 7.42 | 7.47 |
| Spelling | V | 8.29 | 12.77 | 12.75 | 3.25 | 5.46 | 5.42 |
| | IV | 38.33 | 32.08 | 32.13 | 24.74 | 20.47 | 20.65 |
| | III | 39.27 | 37.46 | 37.50 | 42.88 | 38.92 | 39.18 |
| | II | 11.01 | 12.25 | 12.19 | 20.77 | 21.82 | 21.74 |
| | Ib | 2.77 | 4.16 | 4.14 | 7.14 | 9.78 | 9.57 |
| | Ia | 0.33 | 1.28 | 1.30 | 1.23 | 3.54 | 3.44 |
| Mathematics | V | 4.45 | 6.17 | 6.11 | 6.66 | 9.05 | 8.84 |
| | IV | 16.62 | 15.83 | 15.84 | 22.10 | 20.66 | 20.14 |
| | III | 31.58 | 31.36 | 31.63 | 32.13 | 33.29 | 32.46 |
| | II | 32.17 | 30.19 | 30.02 | 26.81 | 24.25 | 24.84 |
| | Ib | 12.58 | 12.80 | 12.69 | 9.63 | 9.38 | 10.01 |
| | Ia | 2.60 | 3.65 | 3.72 | 2.67 | 3.37 | 3.70 |

When the proportions of the students at each proficiency level were evaluated, one of the important results was that in the reading and spelling tests, more female students were classified at high proficiency levels (IV, V) than male students. The proportion of the male students who were classified at low proficiency levels was also higher than that of the female students; for instance, for the reading test, at the highest proficiency level (V) using the Rasch model, the proportion of the female group was 11.32%, and the proportion of the male group was 6.71%. Similarly, for the spelling test, 8.30% of the female students were classified at the highest proficiency level, while this value was 3.25% for the male group. In contrast to the reading and spelling tests, for the mathematics test, the proportion of the male students was higher than that of the female students. The results showed that for the 2PL and PI-2PL models, the proportion of students at the lowest two proficiency levels (Ia, Ib) and the highest proficiency level (V) was greater than for the Rasch model. The proportion of students at the other proficiency levels (II, III, IV) was the same or lower than the Rasch model classifications. These results showed that the effects of the misfit items were greater on the lowest and highest proficiency levels than on the middle levels.

**Table 8.** Proportion (%) of students at each proficiency level, by language subgroup.

| Tests | Proficiency Levels | German | | | Non-German | | |
|---|---|---|---|---|---|---|---|
| | | Rasch | 2PL | PI-2PL | Rasch | 2PL | PI-2PL |
| Reading | V | 10.42 | 13.89 | 13.90 | 1.74 | 2.67 | 2.88 |
| | IV | 26.13 | 24.12 | 24.11 | 11.07 | 10.68 | 10.87 |
| | III | 31.06 | 29.58 | 29.55 | 24.33 | 23.12 | 22.71 |
| | II | 24.47 | 20.48 | 20.52 | 35.20 | 26.92 | 26.62 |
| | Ib | 5.50 | 7.85 | 7.86 | 16.35 | 18.55 | 18.35 |
| | Ia | 2.42 | 4.08 | 4.07 | 11.32 | 18.07 | 18.58 |
| Spelling | V | 8.29 | 12.66 | 12.64 | 3.01 | 4.87 | 4.83 |
| | IV | 35.94 | 32.46 | 32.41 | 22.28 | 20.39 | 20.15 |
| | III | 41.38 | 36.89 | 36.95 | 44.19 | 38.28 | 38.50 |
| | II | 11.46 | 12.50 | 12.54 | 21.18 | 22.10 | 22.04 |
| | Ib | 2.59 | 4.35 | 4.33 | 7.67 | 10.03 | 10.27 |
| | Ia | 0.34 | 1.14 | 1.13 | 1.67 | 4.33 | 4.23 |
| Mathematics | V | 4.67 | 6.14 | 6.12 | 0.96 | 1.28 | 1.29 |
| | IV | 15.15 | 16.09 | 16.11 | 5.37 | 6.07 | 5.80 |
| | III | 36.42 | 31.64 | 31.69 | 22.61 | 19.38 | 19.47 |
| | II | 28.02 | 29.70 | 29.69 | 30.76 | 31.65 | 31.84 |
| | Ib | 12.72 | 12.81 | 12.75 | 25.77 | 25.46 | 25.67 |
| | Ia | 3.03 | 3.62 | 3.64 | 14.53 | 16.15 | 15.93 |

Similarly, we compared the proportions of students at different proficiency levels with the Rasch, 2PL model, and PI-2PL model for the language subgroups. The results for the proportions of students in the language subgroups are presented in Table 8. The findings indicated that for all models, the proportion of the non-German students at the highest proficiency levels was smaller than the proportion of the German students, and the proportion of the non-German students was higher at the lower proficiency levels than the proportion of the German students. German students had the highest proportion at the highest proficiency level on the reading test, and the lowest on the mathematics test. Non-German students had the highest value on the highest proficiency level on the spelling test. Furthermore, because there were not more misfit items with the 2PL model, the classifications based on 2PL and PI-2PL were very similar for both subgroups.

After investigating misfit items and classifications of the students in terms of their proficiency levels, we evaluated the internal reliability of the test scores for all models. These results are presented in Figures 4 and 5. Reliability was computed for gender and language groups, and these values were compared based on the Rasch, 2PL, and PI-2PL models. The reliability of the reading test for gender groups was 0.855 (Rasch), 0.863 (2PL), and 0.864 (PI-2PL). According to the results, the PI-2PL and 2PL models provided higher reliability values than the Rasch model for every test. The reliability of the German test using the PI-2PL model was the highest (0.925); however, the values that were estimated based on the 2PL and PI-2PL models were very close to one another. Because the numbers of the items on the German test ($n = 103$) and the spelling test ($n = 62$) were greater than those for the reading ($n = 41$) and mathematics tests ($n = 48$), the internal reliability of the German and spelling tests was also greater than that of the reading and mathematics tests. The reliability values for the reading test with language misfit were 0.857, 0.865, and 0.867 for the Rasch, 2PL, and PI-2PL models, respectively. The Rasch model provided the lowest reliability values. Moreover, similar reliability values were computed with the 2PL and PI-2PL models. Because 0.70 was used as a cut value, the reliability estimated with all models was greater than 0.70, and the reliability values of all tests with each model were considered acceptable.

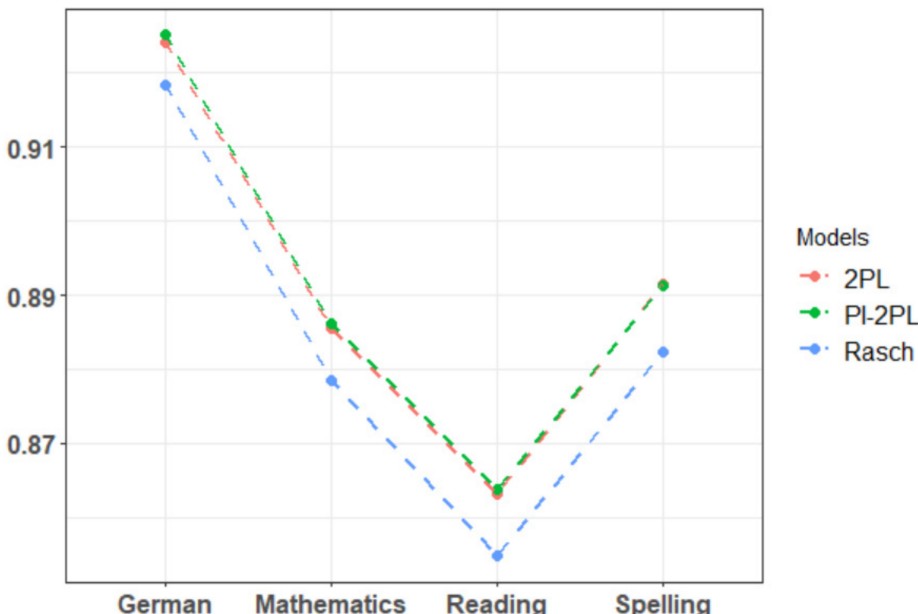

**Figure 4.** Reliability of test scores with multigroup concurrent calibration on gender misfit.

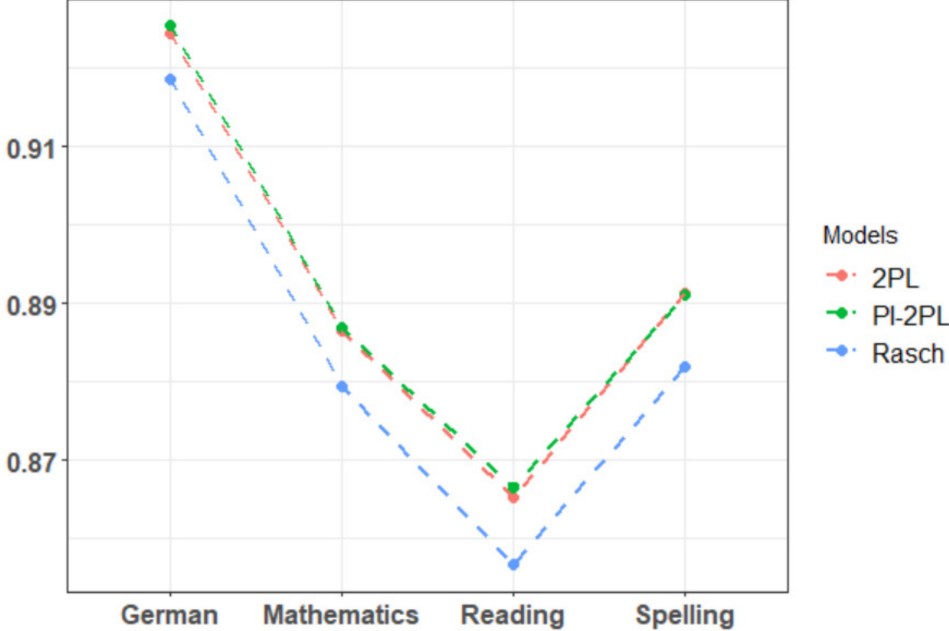

**Figure 5.** Reliability of test scores with multigroup concurrent calibration on language misfit.

## 4. Discussion

The results of this study are consistent with those of some studies in the literature [11,20], in that linking methods may yield less biased items when the DIF effect is balanced. We showed that the mean RMSD values were almost the same and close to 0 in all tests for both subgroups, and that 2PL with full invariance also yielded a very low proportion of misfit items. Furthermore, we did not find any significant effects of removing items with partial invariance on student achievement levels. The proportion of students in the high achievement levels was similar or lower when partial invariance was used. On the other hand, removing these inappropriate items may lead to underrepresentation of constructs [17,18], and one of the suggestions in the literature [17,18] was that the items should be accompanied by expert ratings of the items that exhibit DIF. The estimated reliability of the models was also very similar.

The purpose of this study was to investigate whether the observed gender and language group differences in VERA scores could be explained by a substantial amount of gender- and language-specific DIF. In the study, we found gender- and language-specific DIF with the Rasch model on all tests, and several items had a large misfit (RMSD $\geq$ 0.1) between German and non-German students. Except for the reading test, some items had a large misfit (RMSD $\geq$ 0.1) between female and male students. When we analyzed the proportion of items with medium or large misfit (RMSD > 0.05), female students had more misfits than male students, and non-German students had more misfits than German students. If we evaluated only the items with large misfits (RMSD > 0.1), there were two items with large misfits for both the spelling and German tests for male students; in contrast, there was one item with a large misfit for female students on these tests. However, for both the medium and large misfit items, non-German students had more misfits than German students. In general, most of the items showed a medium or large misfit for both gender and language groups with the Rasch model, and medium or small misfits with 2PL (with the equal item parameters for misfit items). The items showed a small or negligible misfit with PI-2PL (with unequal estimates of these items across groups). However, when 2PL was used, this amount was smaller than with the Rasch model for both language and gender groups. For example, when we used the 2PL model, there were only two items in the reading test for both focus groups (female and non-German students) that had a medium or large misfit. The spelling test included any items for female students and six items (medium or large misfit) for non-German students, and there were five items (medium or large misfit) for both female and non-German students in the mathematics test. The German test included 2 items (medium or large misfit) for female students and 10 items for non-German students.

Because a unidimensional Rasch model was used in the application of VERA, the results of this study provide important implications for VERA applications. One of these is the use of the Rasch model. According to the results of the study, when the Rasch model was used for estimation, more misfit items were found for all subgroups, and it was clearly found in the study that using the 2PL model removed these misfit items for all groups. When evaluating the German and non-German students, another important finding was the unbalanced sample size in the tests. The sample size of the German group was much larger than that of the non-German group. With this, without weighted sampling, we detected more misfit items than with the weighted sample of the German and non-German students. Specifically, misfit items were also detected for the female and male groups with 2PL in the mathematics test, and using PI-2PL models we removed these misfit items from the tests. It should also be noted that we found that there were no large differences between German and non-German students when we applied the Rasch model with weighted sample groups. The literature also highlights that model uncertainty is relatively small compared to sampling error. It was suggested in one study [19] that, in addition to the tendency to choose the better-fitting model, the researcher should also pay attention to whether this choice is appropriate for the research context. We argue that the model error should be included in the statistical inference of large-scale educational assessment studies.

Misfit item values between gender groups were generally not very different, but they were considerably different between German and non-German students. Non-German students had the most misfit items on each test. According to the results that we obtained, language-specific DIF was greater than gender-specific DIF on the VERA tests without sampling weights. Consistent with previous studies [35,71,72]), female students performed better than male students on the reading and spelling tests, and in contrast to these tests, male students performed better than female students on the mathematics tests. The pattern was different for the language groups; on the overall tests, German students were more successful than non-German students, and non-German students tended to be classified at the lower achievement levels.

In addition to examining the misfit items, we examined the effects of the misfit items on the proficiency levels. Two important points can be extrapolated from these results: One

is that when a better-fitting model was used (when using the 2PL model instead of the Rasch model), more students were classified at the highest achievement level for all groups. The second is that because more misfit items were not detected with the 2PL model, the classifications with unequal item parameters from the 2PL model (PI-2PL) were identical to the classifications from the 2PL model. In other words, the items with smaller RMSD values (i.e., small or negligible misfit) did not lead to differences in performance classifications, whereas the items with large misfit led to differences in proficiency classifications.

The possible sources of the gender and/or language differences have been investigated in numerous studies [73–75] to explain the language or gender differences in the tests. Possible sources of gender differences in PISA literacy include, for example, sociocultural and biocognitive factors, as well as test-taking behaviors [35]. The possible sources of the language gap in the VERA test showed a similar pattern to that of international large-scale tests. For non-German students who do not speak German all day, there could be different and specific sources for VERA tests. For example, because these students have less opportunity to speak German at home, they may not understand the German text as well as German students [28]. Other sources could be related to lower socioeconomic status, as the non-German students coming from immigrant families [29]. However, since the students were classified into different proficiency levels, the differences between the male, female, German, and non-German students could not be explained by this small bias of the test items. The reasons for the differences between genders and language groups in the reading, spelling, and mathematics tests (e.g., text sources, text formats, cognitive processes, or students' attitudes) should be examined. The sources of differences between German and non-German students should be clarified by diagnostic or explanatory studies.

In this study, we used a similar pattern as in the original VERA applications. Since every student had a test score in the VERA process, we used the same total sample size as the original applications. On the other hand, there were important findings about the effects of test conditions on RMSD estimation. It is known from the literature that RMSD values depend on the sample size (i.e., when the sample size increases, the RMSD decreases) and the number of items (i.e., more items lead to higher RMSD values) [37]. It is also known that the RMSD is sensitive to the amount of intergroup variability [36]. We evaluated different empirical datasets, but the measurement invariance of tests with different sample sizes should be evaluated (e.g., with random sampling procedures or in k-fold cross-validation studies).

**Supplementary Materials:** The following supporting information can be downloaded at: https://www.mdpi.com/article/10.3390/psych4030030/s1. Figure S1: Item probability functions of reading test items across language groups; Figure S2: Item probability functions of reading test items across gender groups.

**Author Contributions:** G.Y.T. developed the original idea and study design, conducted the data analyses, and wrote most of the overall study; C.R. provided comments on the research process and contributed to editing; M.M. made an important contribution to official permission for the VERA datasets, wrote some parts of the VERA assessments, contributed to editing, and provided comments on the study; R.B. contributed to editing and provided comments on the study. All authors have read and agreed to the published version of the manuscript.

**Funding:** This research was funded by the Ministry of Science, Research, and the Arts Baden-Württemberg with the BRIGITTE-SCHLIEBEN-LANGE-PROGRAMM and the Heidelberg University of Education within the framework of internal research funding.

**Informed Consent Statement:** Not applicable.

**Data Availability Statement:** The datasets analyzed for this study cannot be made available because they are official datasets that require approval from the "Institute for Educational Analyses Baden-Württemberg (IBBW)" at poststelle@ibbw.kv.bwl.de and the "Institute for Quality Development in Education (IQB)" at https://www.iqb.hu-berlin.de/institut/staff (accessed 8 June 2022).

**Acknowledgments:** We would like to thank Nina Jude for her wonderful suggestions.

**Conflicts of Interest:** The authors declare no conflict of interest.

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
