# Peer review of "Examining and Improving the Gender and Language DIF in the VERA 8 Tests"

_psych, doi:10.3390/psych4030030_

Round 1

Reviewer 1 Report

I think the manuscript is very confusing - mostly because of language and writing problems.

  1. In their introduction, the authors emphasized the importance of measurement invariance. Generally, all measures should be tested and validated across multiple populations. A native reader of VERA, however, would appreciate more context - why should the measurement invariance of VERA be tested? There are not always invariant items in psychometric tests as is "typical". Perhaps the authors should focus on some contextual arguments - was there any controversy in VERA? Why it needs to be tested, especially since it seems widely used and validated in various studies. Did the measurement issues contribute to gender differences in past studies?

  2. The authors chose to focus on gender and language as the main groups. The authors should identify research gaps and formulate research questions. What about location, SES (although the authors very briefly discuss it in the discussion), immigration/citizenship, school … 

  3. Sample/method - why 8th grade (VERA 8)? Who were the students? How were they collected? More demographic information is needed. How was VERA tested? 

  4. It seems that the authors tested reading (41 items), spelling (62 items) and math (48 items). Please provide some examples. Was there any other subscale in VERA? 

  5. Results - I was not familiar with the models.I hope other reviewers can provide some useful suggestions. The statistics procedure and results sections in the current manuscript are very statistically oriented. I recommend the authors move some statistics and procedures to an appendix or supplementary file and present only the main findings with proper interpretations. Other readers who are not statisticians can also understand the main findings.

  6. There is a lot of repetition with the results in the first three paragraphs. I am unable to comprehend what the authors intended with VERA in Germany due to the lack of background information. Other variables (SES, school, location) are still more interesting to me than gender and language.

Author Response

Dear Reviewer,

Thank you for your comments on our study. We have followed your suggestions and revised the manuscript and also added the Word file with our answers. The certificate of English proofreading can be viewed here. 
Kind regards,

Reviewer 2 Report

The submitted manuscript (ms) investigates differential item functioning (DIF) for the German VERA test. In general, the topic of the ms is of interest. The authors should better motivate their goals of the study.
Detailed comments:
1.    General observation: I do not really like the presentation style in the Results section. There is a lot of prose (which is not bad if it transmits explanations and insights), but almost no concrete statistics were presented that can be used to compare with other studies. Also, I wouldn't say I like the extensive use of (colored) figures. I always prefer presenting the relevant information in text or tables because numbers cannot be simply extracted from figures. If all results are hidden in figures, it indicates a non-replicable study. I do not argue that one should abandon figures at all (I find the RMSD histograms in the supplement useful) if one gets a quick overview. But figures should always only accompany numerical findings, and they should not replace the presentation of numerical findings. I think that the authors should completely restructure the Results section in a resubmission.
2.    Abstract: I am unsure whether one needs the information “Booklet 1” in the abstract because no information about this booklet type is presented.
3.    Throughout the ms: write “multiple group” instead of “multigroup”
4.    L. 23: write “RMSD” and “DIF” as keywords and remove “RMDF_DIF”
5.    L. 26: I strongly disagree that measurement invariance is an important aspect of fairness. It is neither necessary nor sufficient for obtaining a fair test (Camilli, 2006; Robitzsch & Lüdtke, 2020). Similar statements run through the entire manuscript. I do not see any justification for such statements, although I am aware of the fact that there is a lot of literature that only exists due to their unjustified claim of the importance of measurement invariance.
6.    L. 32: “measurement comparability”. Same here. Why should invariance be related to comparability? I do not think there is necessarily a relation. There can be noninvariant measures, but this does not invalidate the comparability of measures across groups.
7.    Throughout the ms: Authors frequently use the phrase “relatively new”. Please avoid it. Frequently (like for partial invariance or the RMSD statistic), the approaches are not new. Second, the reference year indicates whether it is new or not. If the paper was read 20 years ago, it sounds a bit awkward to read about a “new method”.
8.    L. 74: remove “diverse”.
9.    L. 78: It is just a claim that the partial invariance (PI) approach ensures meaningful comparisons under noninvariance. However, any method can be justified that fulfills appropriate identification constraints, and there is no reason to believe that PI has a special role (Robitzsch & Lüdtke, 2020; Robitzsch, 2022).
10.    L. 100: The large parenthesis in the formula must open after the integral sign.
11.    L. 116: remove “:”
12.    L. 45, 117: I do not agree that VERA is a low-stakes assessment if it is administered by school-internal teachers. Certainly, it is less low-stakes than PISA.
13.    L. 125: I do not know whether the information about the solution probability is relevant. I thought that this information was only relevant for standard setting.
14.    L. 173: It is said that abilities are biased if the model does not fit. In my opinion, this is an incorrect statement. Bias is unrelated to model fit. A model is chosen to define a statistical parameter of interest. Such a definition is entirely independent of model fit aspects. Moreover, a fitting model might output "unbiased" parameters in your definition but are entirely unrelated to parameters of interest. Only external criteria (i.e., external validity) could help select a model among plausible alternatives.
15.    L. 176: Note that information criteria are only relative model fit indices.
16.    L. 211ff.: Note that the PI approach implies that comparisons between groups are only conducted on the items that are declared to be invariant. Hence, in essence, not all items are used for group comparisons (Robitzsch & Lüdtke, 2020). The ms does not highlight this issue. This property is why the mechanistic application of PI does not guarantee meaningful (or valid) comparisons of groups.
17.    L. 224: Provide references for all technical terms and R in this part.
18.    L. 238: write “local dependence”?
19.    L. 230: Provide concrete numerical findings of model fit and test statistics for local independence.
20.    I do not like the abbreviation “R-2PL” for rescaled 2PL model. I would prefer “PI-2PL”.
21.    Fig. 1, Fig. 2: These figures illustrate the issue when applying the RMSD item fit statistic. In the PISA applications, they are applied in the case of many groups; that is, the assessment of fit refers to international item parameters. For two groups, the fitted item response function (IRF) is a weighted summary of the group-specific IRFs. It is clear for the language DIF comparisons that estimated item parameters are primarily based on the German-speaking students. Hence, it is clear that the items show more frequently DIF for NA German students than DIF for German students. However, in the two-group case DIF assessment, a symmetric treatment of groups might be advantageous (Wainer, 1993). In this setup, DIF would be defined as the integrated squared difference between the IRFs of the two groups. As an alternative to using the less appropriately utilized RMSD statistic in your application, one can reweight the data such that the groups have an equal total sampling weight (e.g., 50% of the weighted sample) to let the fitted IRF equally depend on both groups. Then, one applies the RMSD to the weighted sample. In such an approach, this modified RMSD statistic would appear more similar between the two groups. The issue is more relevant for the language DIF than for the gender DIF comparison because the group proportions were more similar in the latter case.
22.    Fig. 3 to 6: The ordering of values of the x-axis for the items is of no relevance. Please include histograms for the RMSD statistic. Use the histograms from the supplement.
23.    Fig. 7 to 9: Include one digit after the decimal. Replace the figures with tables. Please arrange the models beside each other to allow an easier comparison of the different models.
24.    P. 12: Report reliabilities as numbers in the text.
25.    L. 434: I do not see why “the biased items are improved” by imposing the partial invariance assumption. Why should the items be biased? Consult the framework of construct-relevant and construct-irrelevant DIF (see the references in Robitzsch & Lüdtke, 2020).
26.    L. 436: why does the presence of DIF lead to invalid comparisons? In contrast, removing items from linking (e.g., by imposing the PI assumption) leads to invalid comparisons of groups. The ms is full of claims without any validity evidence.
27.    Throughout the ms: “NA German students” sounds quite strange. Maybe better to use “non-German” or “non-German speaking students”?
28.    L. 475ff.: I do not see why a large number of students or items should be a limitation. Moreover, unbalanced group sizes are not generally a limitation of the study.
29.    Obviously, the authors (as well as many others in the literature) prefer the PI assumption of DIF effects. As with any other assumption in statistical analysis, I would like to see a discussion on whether there are only a few large outlying DIF effects, while most of the DIF effects should be zero or small. In my experience and in accordance with your RMSD plots, I think DIF effects are rather symmetrically and normally distributed. At least, the preference of PI should be justified based on empirical evidence of this type of DIF effects in your dataset.
30.    Supplement: include Table 1 in the main text
31.    Supplement: Fig. 1 has no empirical meaning. Please remove it.

References:
Camilli, G. (2006). Test fairness. In R. L. Brennan (Eds.), Educational measurement (pp. 221–256). Westport: Praeger Publisher.

Robitzsch, A. (2022). Estimation methods of the multiple-group one-dimensional factor model: Implied identification constraints in the violation of measurement invariance. Axioms, 11(3), 119.

Robitzsch, A., & Lüdtke, O. (2020). A review of different scaling approaches under full invariance, partial invariance, and noninvariance for cross-sectional country comparisons in large-scale assessments. Psychological Test and Assessment Modeling, 62(2), 233-279.

Wainer, H. (1993). Model-based standardized measurement of an item's differential impact. In P. W. Holland & H. Wainer (Eds.), Differential item functioning (pp. 123–135). Lawrence Erlbaum.

Author Response

Dear Reviewer,
Thank you very much for all your suggestions. Your suggestions were really a very important contribution to our study. We are very grateful to you for them. We have followed all your suggestions, which could be found in the new manuscript with Track Changes” function.

Kind Regards,

Reviewer 3 Report

This study caught my attention because it focuses on measuring psychometric qualities, respectively identifying misfit items of the test (reading, spelling, mathematics) used nationwide in schools in Germany. It was based on advanced analyzes in terms of IRT approach, more precisely on a new method, i.e., RMSD.

The sample is very large but it is not mentioned if it is the representative sample

The assumptions required for IRT analysis were followed.

You must explain (at line 225) why you chose the respective estimation methods, ie MML and WLE.

It would be desirable to specify the recommended cut off for empirical reliability.

Explain the sampling method, more precisely why you chose Booklet 1 which is recommended for low achievers students

It is not clear to the reader why you chose to investigate only low-achieving students if the differences in gender and language are explained and to what extent the bias of the items.

Mention in line 472 that a possible source of the differences investigated would be related to cultural differences. There is a whole literature on this topic that shows that such sources can be advantages for some and disadvantages for others in assessment contexts.

Author Response

Dear Reviewer,

Thank you for your comments on our study. We have followed your suggestions and revised the manuscript and also added the Word file with our answers.

Kind Regards,

Round 2

Reviewer 1 Report

The authors revised the manuscript responsively. I don't have other comments. Well done!

Author Response

Dear Reviewer, 

Thanks again for all your comments.

Regards,

Reviewer 2 Report

The revised manuscript (ms) investigates differential item functioning (DIF) for the German VERA test. I agree with most of the changes. I have a few remaining comments:
1.    General comment: If reviewers provide a list with additional manuscript-related references (Camilli, 2006; Robitzsch, 2022a; Robitzsch & Lüdtke, 2020), it is standard, in my opinion, to implement them in the ms. If the authors decide not to include a suggested reference, they should argue why they don’t in the response letter.
2.    According to the authors' response, they believe that according to APA testing standards DIF would be a threat of validity because “validity refers to the degree to which evidence and theory support the interpretations of test scores for proposed uses of tests”. I think that this is a misread of the APA standards. The authors should search for any place in the APA standards that DIF items must be removed or DIF is necessarily related to test fairness (see also Camilli, 2006; which was suggested by me in the first review but not implemented in the manuscript). I also read the APA standards a while ago, and I am certain that you cannot find arguments that comparisons will automatically be invalid if there is DIF (see, e.g., the section about test fairness and the book chapter of Camilli, 2006).
3.    If the authors observe that applying the partial invariance (PI) model would not change results, what is the motivation for using PI? I think that the response that we do not observe practical differences between PI and the DIF-ignoring approach does not mean that one can always use the PI approach. In contrast, my point is that it is inappropriate to state that the PI approach is less biased than any alternative modeling approach (Robitzsch, 2022a). The ms does not mention this relevant point.
4.    L. 654: It is still said that item parameters (and group comparisons) can be biased in case of a misspecified IRT model. Although one can also find such statements in the literature, they are nevertheless nonsensical. My point is that model parameters can also make sense and are very informative in the case of misspecified models (Berk et al., 2014; Boos & Stefanski, 2013, Ch. 7, https://doi.org/10.1007/978-1-4614-4818-1_7; Robitzsch, 2022b; White, 1982).
5.    RMSD computation: I do not see why subsampling of students is required. One can always use sampling weights to avoid introducing more sampling uncertainty. For example, if group A has 200 students in the sample and group B has 1000 students, the sampling weights for students in group A would be 1.0, and in group B, they are 200/1000=0.2.
6.    Comment and response 26 of the previous review: I questioned the statement that the presence of DIF implies invalid comparisons. In contrast, I argue that the PI assumption does not resolve this issue (Robitzsch & Lüdtke, 2020). The authors responded that they did not observe substantial differences among methods, and most of the item parameters could be assumed to be invariant. The response is unrelated to my critique. Why should one use PI methods if DIF appears only in a few items? I also think that this observation does not answer the question about the adequate modeling approach. Authors should ask which method would be preferred if there is some difference between the PI and the full invariance approach.
7.    Comment 29 of the previous review: Coincidentally (?), there was no response to my point that I want to see the PI assumption of DIF effects being supported in the data. I asked you to provide empirical evidence for your dataset. At least, the point of distributional assumptions of DIF effects should appear elsewhere in the ms because the distribution of DIF effects implies which linking strategies should be utilized (Robitzsch, 2021).
8.    L. 31: It is stated that “according to IRT”, lack of measurement equivalence (ME) can be detected. It seems that ME is used synonymously with DIF. If this is not intended and ME is meant to be synonymous with comparability, the statement is wrong; IRT has nothing to say about ME.
9.    L. 143: Is “item bias” meant equivalent to DIF? If yes, please avoid the wording “item bias” here. In my opinion, item bias is a different concept than DIF, and DIF is neither necessary nor sufficient for item bias.
10.    L. 562: write “Rasch model” instead of “IRT Rasch model”
11.    L. 741ff.: please include labels in the parenthesis for fit statistics for the two tests, e.g., “TLI (reading: .923; mathematics: .943)” and do the same for other statistics.
12.    L. 748: typo “% 14.52”
13.    Discussion: Please add some discussion about the discussion 1PL vs. 2PL (see, e.g., Robitzsch, 2022, and the references therein).

Relevant references to be included in the manuscript:
Berk, R., Brown, L., Buja, A., George, E., Pitkin, E., Zhang, K., & Zhao, L. (2014). Misspecified mean function regression: Making good use of regression models that are wrong. Sociological Methods & Research, 43(3), 422-451.

Boos, Dennis D., and Leonard A. Stefanski. Essential statistical inference. Springer, 2013.

Camilli, G. (2006). Test fairness. In R. L. Brennan (Eds.), Educational measurement (pp. 221–256). Westport: Praeger Publisher.
Robitzsch, A. (2021). Robust and nonrobust linking of two groups for the Rasch model with balanced and unbalanced random DIF: A comparative simulation study and the simultaneous assessment of standard errors and linking errors with resampling techniques. Symmetry, 13(11), 2198.
Robitzsch, A. (2022a). Estimation methods of the multiple-group one-dimensional factor model: Implied identification constraints in the violation of measurement invariance. Axioms, 11(3), 119.
Robitzsch, A. (2022b). On the choice of the item response model for scaling PISA data: Model selection based on information criteria and quantifying model uncertainty. Entropy, 24(6), 760.
Robitzsch, A., & Lüdtke, O. (2020). A review of different scaling approaches under full invariance, partial invariance, and noninvariance for cross-sectional country comparisons in large-scale assessments. Psychological Test and Assessment Modeling, 62(2), 233-279.
White, H. (1982). Maximum likelihood estimation of misspecified models. Econometrica.

Author Response

Dear Reviewer 2,

I am very appreciative for your comments and apologize for the first round where I was not able to include all the references you suggested. I have read them all and included them in the manuscript. I used the relevant arguments in these studies, not only for this manuscript, but in general I learned more from all the discussions. You could also find two new figures and a table that you asked about the point of distributional assumptions of DIF effects. I also used weighted sampling and re-estimated the RMSD values. 
Please find the answers attached. 
Regards,

Round 3

Reviewer 2 Report

I am satisfied with the revised version of the manuscript.